# Stop overkilling simple tasks with black-box models, use more transparent models instead

## Abstract

The ability of deep learning-based approaches to extract features autonomously from raw data while outperforming traditional methods has led to several breakthroughs in artificial intelligence. However, it is well-known that deep learning models suffer from an intrinsic opacity, making it difficult to explain why they produce specific predictions. This is problematic not only because it hinders debugging but, most importantly, because it negatively affects the perceived trustworthiness of the systems. What is often overlooked is that many relatively simple tasks can be solved efficiently and effectively with data processing strategies paired with traditional models that are inherently more transparent. This work highlights the frequently neglected perspective of using knowledge-based and explainability-driven problem-solving in ML. To support our guidelines, we propose a simple strategy for solving the task of classifying the ripeness of banana crates. This is done by planning explainability and model design together. We showcase how the task can be solved using opaque deep learning models and more transparent strategies. Notably, there is a minimal loss of accuracy but a significant gain in explainability, which is truthful to the model's inner workings. Additionally, we perform a user study to evaluate the perception of explainability by end users and discuss our findings.

## 1 Introduction

Over the last decade, Machine Learning (ML) research has been increasingly focused on developing new deep models based on Artificial Neural Networks (ANNs). Such methods have raised the bar in accuracy for numerous cognitive tasks, leading to new and exciting opportunities and serious challenges. Among these challenges, *explainability* has sparked a vast amount of discourse and debate; briefly, it can be understood as the endogenous process of communicating information about the model and data to foster human understanding of the decision-making process of such models (Rizzo et al., 2022). This is no trivial task, especially for modern deep models, as these are highly complex and rely upon billions of opaque parameters that must be learned during training.

Deep Learning (DL) models' incredible performance paired with their inner opacity constitute a concrete problem. This is because explainability is crucial to verify the model's properties, such as fairness and trustworthiness, especially in high-stakes decision-making environments. With the ever-growing use of AI in many application fields, policymakers are supporting the need for explainability (Selbst & Powles, 2017). In Europe, for example, a significant effort to regulate models' decisions by endorsing the user's right to an explanation is being made in the writing of the AI Act (EU, 2021). Unfortunately, it is clear how this clashes with much of the design process of DL models, which is generally guided by researchers' intuition, relies on trial and error for tuning and lacks a holistic approach, including the upstream definition of an explanation strategy. Recently, a theoretical framework proposed by Rizzo et al. (2022) has tried to provide common ground for the meaning of keywords used with no real shared purpose in the XAI community. We resort to this framework for our backing definitions of the terms above. To briefly summarise these notions, an *explanation* is an answer to a why question derived from *interpreting* some *evidence* (*i.e.*, factual information) This splits the concept of explanation into two new atomic components, the latter speaking of how much of the model we can explain (*i.e.*, how much our evidence is involved in the model's computation) and the former telling how the evidence is transformed inside the model. Interpretations are

Figure 1: Ripeness stages for crates of bananas from least ripe (1) to ripest (4).

hypotheses; thus, they should be tested for faithfulness (*i.e.*, is the interpretation capturing what the model is doing?) and plausibility (*i.e.*, does the interpretation align with the stakeholders' intuition of how the model works?). Lastly, the information content of an explanation should be presented to the user through an eXplanation User Interface (XUI) (*i.e.*, text, plots, interactive interfaces, etc.) to verify the effectiveness of the knowledge transfer (Rizzo et al., 2022).

A broad spectrum of methods is now designed for minor accuracy improvements over their predecessors. Unfortunately, the speed at which these are being developed far outpaces the development of strategies capable of explaining them. Research towards explainability methods has nevertheless brought exciting results, with milestone techniques such as SHAP (SHapley Additive exPlanations) (Lundberg & Lee, 2017). Such a method attempts to explain the prediction of any classifier by using an approach grounded in game theory. While still widely used, it has received criticism, as it may provide explanations that are, at the very least, disputable (Kumar et al., 2020; Alvarez-Melis & Jaakkola, 2018). Unfortunately, using one-fit-all methods such as SHAP does not work around the need for a better overall understanding of the designed solution. In this paper, we highlight an approach to problem-solving in ML that draws from often-forgotten simple ML models and applies a modern all-around design and analysis of model explainability. We showcase our strategy with a simple, but hopefully very clear, practical example relevant to the industry.

## 1.1 TASK AND APPROACH

To showcase our design strategy, we analyse a straightforward real-world scenario and how the aforementioned concepts of accuracy and explainability affect it. Our target task is the classification of the ripeness of banana crates on a scale from 1 (least ripe) to 4 (ripest) (see Fig. 1 for an example).

In our approach, we design for competitive accuracy and explainability simultaneously. To tackle the classification task, we select a pool of three DL methods: (i) a simple Convolutional Neural Network (CNN) model with three convolutional blocks, (ii) a pre-trained convolutional model based on the MobileNetV2 framework (Sandler et al., 2018), and (iii) a pre-trained Vision Transformer (ViT) (Dosovitskiy et al., 2020). As we will show, the latter allows for almost perfect results and is the best neural model across our proposed methods, but at the cost of no current method capable of explaining its prediction accurately. On the other hand, we show that our approach, based on simple colour features and a fine-tuned Decision Tree (DT), can provide competitive accuracy while exposing the information needed for producing adequate and global explanations.

## 1.2 CONTRIBUTIONS

Our experiments show that all three selected neural models can converge to very high (and, in some cases, close-to-perfect) accuracy in a few training epochs. This leads us to question the difficulty of the task at hand and the actual need for such powerful yet black-box methods. As the results suggest, the task is somewhat easy, and it is thus legitimate to tackle it with a simpler strategy. The expectation is that, despite the simplicity, the new model will be able to reach competitive accuracy while leaving us space for integrating explainability into our design.

In summary, our contributions are the following:

- We provide high-level design guidelines to tackle ML problems with explainability in mind;
- We showcase an explicative classification task, for which we provide an analysis of a selection of DL methods in terms of accuracy and explainability, utilising relevant models that offer a wide panoramic of the task;

- Moreover, we show that the same classification task can be solved effectively and efficiently by a much simpler and more transparent model, a DT, with minimum feature engineering effort;

- We conduct a user study to determine which explanations best suit the stakeholders' needs;

- We release our code and self-collected dataset[1] for reproducibility and possible extendibility of our experiments.

## 2 RELATED WORK

Our work relates to two main paths of research: (i) the advocacy for more focus on explainability in the Artificial Intelligence (AI) community and (ii) the optimisation of fruit ripeness grading. We proceed to briefly introduce previous works on these subjects.

**AI explainability.** A common problem associated with DL models is their inner opacity. Providing meaningful explanations for a DL model's prediction is an arduous task. Much research has gone towards the extraction of explanations by using, for instance, information from gradients (Selvaraju et al., 2017), attention scores (Bahdanau et al., 2015a), surrogate models (Ribeiro et al., 2016), and latent prototypes (Chen et al., 2019). Some methods have been proposed with the promise of being model-agnostic, *i.e.*, to explain the prediction of any classifier. Prominent examples are the aforementioned LIME and SHAP methods (Ribeiro et al., 2016; Lundberg & Lee, 2017). However, the proposed explanations have been challenged (*e.g.*, (Adebayo et al., 2018; Serrano & Smith, 2019; Garreau & Mardaoui, 2021; Nauta et al., 2021; Khakzar et al., 2022)) and proved to be unreliable in multiple scenarios. Nevertheless, SHAP is still considered state-of-the-art for explainability by many. Moreover, it allows the combination of multiple local explanations to produce a "*global*" (averaged) explanation of the model instead of the *local* explanation of a single prediction. Similarly, our proposed explanation strategy is global. For these reasons, we selected SHAP as our benchmark strategy to explain the DL models and to compare them against our proposed solution.

**Fruit ripeness recognition.** Grading the ripeness of the fruit is a long-studied problem for whom strategies based on statistics (*e.g.*, (Mendoza & Aguilera, 2006; Olarewaju et al., 2016)), traditional ML (*e.g.*, (Ni et al., 2020; Septiarini et al., 2020)), and DL (*e.g.*, (Saranya et al., 2022; Sa et al., 2016)) have been proposed. The top-performing methods are those based on DL, which, aside from reaching astonishing accuracy, do away with the complex and error-prone task of feature engineering. However, the literature lacks extensive comparisons among the three noted strategies. For more information on the fruit ripeness grading problem and solutions, a recent survey was authored by Rizzo et al. (2023).

On another note, the focus of much of the most recent research appears to be on scraping a few decimals of task accuracy (or other performance indicators) while too often not accounting for explainability (Marcuzzo et al., 2022). On this topic, works such as the one by Rudin (2019) discuss the necessity of more carefully gauging the tasks being solved and, whenever possible (or necessary due to high stakes), using more transparent models rather than black boxes. Our thesis is similar: we advocate choosing the most simple and transparent model that achieves a satisfying performance while also devising a strategy to faithfully explain its behaviour.

## 3 DESIGNING FOR EXPLAINABILITY

Our proposed guidelines aim to find the problem features that are more intuitive for the stakeholders and process them as little as possible through the simplest ML method adequate for the task. "Simplicity", in this case, relates to the number of parameters regulating the model (the lower, the better) and its reliance on human-understandable processing of the features (the more, the better).

In particular, we want to produce a pipeline from raw data to prediction, where each step is as transparent as possible. The proposed design process follows these high-level steps: (i) understand the task to be solved by the ML method, the available data, and the stakeholders of the final product;

---

[1]*Will be made available after anonymous review.*

(ii) for each stakeholder, discuss which attributes they consider relevant in solving the task and define which features can be considered part of an explanation; (iii) find a ML model that is powerful enough to process the features but also offers the possibility to extract interesting evidence with a reasonable effort. The evidence must suggest an interpretation that is faithful by design to how the model works and possibly aligns with human intuition for plausibility (Rizzo et al., 2022); (iv) test model performance and effectiveness of the generated explanations: the model should provide competitive accuracy with the state-of-the-art, while also satisfying the expectations of the stakeholders with the produced explanations. We find that a user study is an effective way to get qualitative evidence of the efficacy of the proposed XUI.

Step (ii) is perhaps the most challenging point, especially when very little problem-specific knowledge is available to the stakeholders. In this scenario, a preliminary analysis of the performance of top black-box models can indicate how hard the task is. If the specific task exposes intuitive features that can be leveraged to solve it, a model that tends towards transparency is worth trying. Intuitiveness is critical to optimising the design and reaching a final explanation faithful to the model behaviour and plausible to the human stakeholder. On the other hand, we acknowledge that finding meaningful features or even just effective data representation can be challenging for some tasks. In Natural Language Processing, for example, handcrafting general context-sensitive and human-understandable features is often very difficult or impractical, partly due to the inherent complexity of languages. That is why we advocate reasoning about an ML problem and try a broader explainability-driven approach, especially when the task is simple. For some tasks, simple or explainable solutions may not be there yet. The following sections showcase how we applied such guidelines to our example task.

## 3.1 Task definition, stakeholders, and data

From a practical perspective, this work deals with a multiclass image classification task. Our stakeholders are workers at the wholesale fruit market of the city of Treviso, Italy, who are interested in automating the ripeness grading of banana bunches. Currently, bunches are manually labelled by operators on an increasing ripeness value (1 to 4, least to most ripe, see Fig. 1). All the bananas within a crate are assumed to be in the same ripeness stage. The ML classifier resulting from this work would be used to aid operators in labelling large numbers of incoming crates. Moreover, this is the first step in the process of digitalization of the fruit processing pipeline, from inspection and assessment of fruit quality to online sales. Given the impact of the assessment step on the pricing of fruit, our stakeholders stressed the importance of maintaining transparency in the grading process, to allow human supervision.

To develop the ML solution, we collected an *ad-hoc* dataset comprising 927 images, with a reasonable balance between the four ripeness classes. The dataset was manually labelled by the operators that perform the quality assessment of incoming products. To understand human performance on this classification task, we also asked three operators to re-label a subset of images from the dataset. More technical details on the data are provided in Section 4.1, while the human performance is reported in Section 5.

## 3.2 Feature selection

After consultation with the stakeholders, we determined that colour is the most reliable and intuitive factor in determining the ripeness of banana bunches. Images are encoded using the well-known RGB colour space, a well-known colour model backed up by solid theory based on the human perception of colours. Since colour is the most important feature of our dataset, we process images such as to precisely extract valuable colour information and train our classifier to recognise the ripeness stage by considering such colour. Section 4.1 details how this information is extracted and used in the proposed solution.

## 3.3 On the choice of models

We select both state-of-the-art DL-based methods and simpler, more transparent classifiers for this task. Testing DL models gives us an idea of the best performance that can be achieved, as well as the difficulty of the problem. As stated previously, our objective is to choose the model of the lowest

complexity that achieves adequate performance, as to preserve as much transparency as possible. We selected a DT, a Support Vector Machine (SVM) with different kernels and a multinomial Naive Bayes (NB) classifier as baseline models for comparison, eventually choosing the DT as the best model of the three.

We point out that the DT learns discriminative rules that partition the feature space into sub-spaces corresponding to each target class (*i.e.*, the ripeness stage). By extracting colour information in the RGB space and limiting the number of extracted features we can obtain a *global* explanation mapping each ripeness stage to specific areas of the colour space. We highlight that this explanation is *faithful*, in the sense that it describes the DT "reasoning" process, as well as *plausible*, meaning that it is aligned with the human understanding of the problem. These characteristics make this strategy effective with respect to the point (iii) in our guidelines.

### 3.4 Testing for accuracy and explainability

We compare the performance of the baseline models (DT, SVM, NB) and select the DT to be the best compromise between the complexity and intuitiveness of the explanation that can be derived from it, as discussed in the previous section. The NB classifier achieves lower performance than the DT. Conversely, the SVM with a high-degree polynomial kernel achieved slightly better results (less than 0.5% accuracy and F1-score improvement). However, given the minimal difference in results and considering that the decision boundaries of the SVM are more difficult to understand because of their complexity, the DT appears to be a better choice. The complete results of these tests are reported in the supplementary material. Additionally, we compare the DT with some state-of-the-art DL models that would be the obvious off-the-shelf DL solutions for this task. Results are reported in Section 5, showcasing that the DT achieves competitive performance, and is well above human classification performance.

Finally, we want to assess the efficacy of the generated explanations for our stakeholders. To do this, we conducted a user study to investigate the users' preferences about the generated explanations. More details on the results are provided in Section 5.2 while the complete questionnaire is reported in the Supplemental materials.

## 4 Methods and explanations

### 4.1 Data processing

Our dataset is composed of 927 RGB pictures of crates filled with various bunches of bananas. Images were acquired at a native resolution of 4160 x 3120 pixels using a CZUR Shine Ultra scanner, with an effort to achieve consistent lighting. The dataset is split among classes in a reasonably balanced way. We detail the class distribution in the supplementary material.

Each image was resized to 224 x 224 pixels to obtain a reasonable inference time. This resizing was also chosen as it is standard in many pre-trained models, allowing us to use modern transfer learning approaches easily. The dataset was augmented with random transformations, including rotation, affine transforms, elastic morphology transforms, random location crop, gaussian blur, as well as the erasure of patches of the image and changes in perspective. The latter transformations were applied to account for different angles and accidental occlusions, likely when non-expert users take pictures with a smartphone (which is one potential end use of this classifier). We augmented roughly 50% of the dataset, and the new images were added to the original dataset before training. More details may be found in the supplementary material.

A visual inspection of the dataset reveals that pictures are noisy in that parts of the crate are captured in the overall image (mostly the boundaries of the crate and, sometimes, its bottom). To circumvent this problem, we perform semantic segmentation of the images to filter out the background of banana bunches. Having no manually segmented images, an unsupervised approach was the only feasible way to achieve this. After testing several algorithms, we selected the SLIC algorithm (Achanta et al., 2012) for this task. In our experiments, all methods benefit from including segmentation as a pre-processing step. As such, we only report results on segmented images.

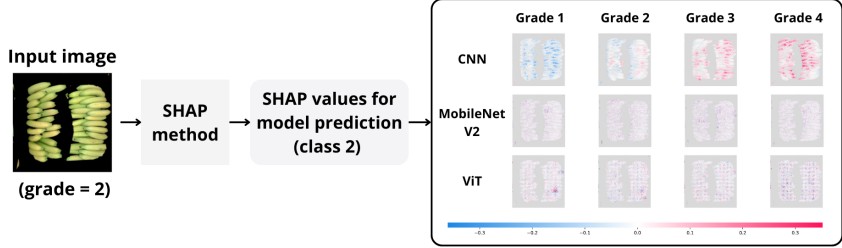

Figure 2: Examples of explanations for DL models generated using SHAP

Moreover, while the selected DL models can automatically extract discriminative features from raw RGB input images, we devised a minimal feature engineering process to extract colour features and use them with the DT. Specifically, each image is represented by three features: the R, G and B channel values of their average colour, normalised in the $[0 - 1]$ range. This way, all the image structure is discarded, leaving only colour information. One notable thing to consider about RGB is how the luminance is embedded within its three channels. This is different, for instance, from colour spaces such as YUV, where luminance is encoded into the physical linear-space brightness (Y) channel. As the DT is based on average colour values, this method benefits from normalising the luminance. This is achieved by transposing all the images to the YUV colour space, setting the Y channel to a common value, and then translating back to RGB.

## 4.2 DEEP LEARNING APPROACH

In addressing the task of banana ripeness classification, we run and compare three neural approaches. The first architecture consists of a simple CNN using three convolutional blocks, each characterised by two bi-dimensional convolutions and max pooling interleaved by ReLU activation functions. The convolutional layers extract features fed to a three-layer feed-forward ANN, which outputs the final prediction. Before being processed by the CNN, the data is normalised to mean and standard deviation.

The second architecture we consider is the pre-trained MobileNetV2 network (Sandler et al., 2018). Still convolutional by nature, the strategy at the core of this method is based on depth-wise convolutions (Sifre & Mallat, 2014; Chollet, 2017) and inverted residual connections. The designers aimed to build a powerful, pre-trainable model for low-tier devices.

The third architecture we examine is the Vision Transformer (ViT) (Dosovitskiy et al., 2020). Transformers (Vaswani et al., 2017) are neural architectures based on multi-head attention (Bahdanau et al., 2015b), widely studied and employed by the NLP community (Gasparetto et al., 2022a;b). This architecture has seen recent applications to CV tasks with various strategies (see (Khan et al., 2022) for a survey). Briefly, ViT splits images into fixed-size patches and linearly embeds them. Positional embeddings are then added to retain position information before feeding the resulting sequence of vectors to a standard Transformer encoder. Classification is achieved by adding a learnable "classification token" to the sequence.

### DEEP LEARNING EXPLAINABILITY STRATEGY

As previously mentioned, we used SHAP (Lundberg & Lee, 2017) to explain the predictions of the DL models. When dealing with images, SHAP allows generating heat maps (which constitute the XUI) to deliver the explanation to the user. These are supposed to describe the importance of each pixel in the image toward the model's prediction. Intuitively, warm colours indicate the regions of the image that contributed the most to the prediction. In contrast, colder colours indicate areas that contributed negatively to the prediction of the same class. Example explanations generated with SHAP are presented in Fig. 2. Previous literature found that SHAP, despite being widely used, produces explanations lacking faithfulness while looking plausible (Rizzo et al., 2022; Alvarez-Melis & Jaakkola, 2018). This is an alarming condition where the explanations convey to the user "a convincing lie" about how the model behaves. The following sections show how our design addresses faithfulness and plausibility.

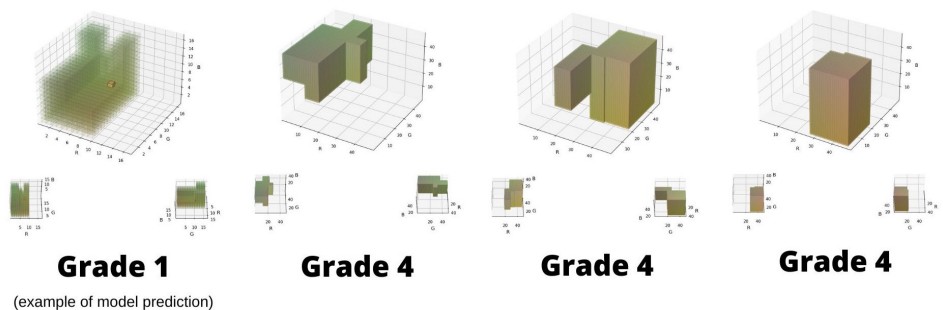

**Grade 1**
(example of model prediction)

**Grade 4**

**Grade 4**

**Grade 4**

Figure 3: Explanation generated from the constraints imposed by the DT on the RGB colour gamut. The four grades identify different areas within the gamut.

### 4.3 DECISION TREE

In contrast to the examined DL methods' inner complexity, we propose tackling the same task using a simple, more transparent model based on a DT classifier. In particular, we adopt the implementation offered by scikit-learn, which is based on the CART algorithm (Breiman, 1984).

#### EXPLAINABILITY STRATEGY

One may argue that a DT is an intrinsically explainable model. We argue that there is no such thing as intrinsic explainability: a transparent model still needs to provide some explanation that is somewhat understandable to the users and answers their "why" questions. Different end-users are likely to have different requirements for explainability. For example, ML experts may be satisfied with understanding the range of feature values mapped to each target class (in our case, the RGB values). Non-expert users may need these rules to be further processed to be represented more clearly. Serving explainability is intuitively much easier with specific models, such as those regulated by a few parameters, though this is yet to be formalised in the literature. Admittedly, a DT has a very intuitive and faithful interpretation: for every non-leaf node, the DT learns a threshold value for one of its given features, thus producing two children (above and below the threshold). In our case, each instance is classified by following a path to a leaf labelled with a specific ripeness value. Conveniently, the set of rules given by the traversed path defines an area within the RGB colour space that is part of our explanation. Binding the explanation to the intuitive process of discriminating banana crates based on colour (as our stakeholders do) sets the premises for it to be plausible.

Albeit simple to follow for relatively shallow trees, the decision paths can grow exponentially for features that have complex interactions. As anticipated, such numerical features split within the DT can still appear opaque to the average user. Thus, we take our explanation further by devising an XUI that aims to be human-understandable and tested accordingly. More specifically, we use the rules extracted from the decision path as constraints on the RGB gamut to identify portions of such a space representing the four ripeness classes. Hence, it is easy to represent each unknown input data point as its average colour in the 3D RGB colour space and determine which region it belongs to. This plot is our proposed explanation for the DT's behaviour. Fig. 3 is an example visualisation of the whole process (more examples are reported in the supplementary material). It's worth stressing that the area of the colour space extracted from the decision rules learned by the DT is, by definition, a *global* explanation. As such, our strategy allows us to unequivocally understand which colours are associated with each label class. One of the benefits of such an interpretable explanation is the ability to validate the classifier's behaviour. Unexpected colours would show up in the proposed XUI, pointing out a negative bias in the model.

## 5 EXPERIMENTS

In this section, we compare the performance achieved by our employed methods. First, we analyse the classification metrics achieved by the three DL-based models and the DT. Then, we study the explanations generated according to the strategies proposed in Section 4.2 and Section 4.3 and compare

|  | Accuracy | Precision | Recall | $F_1$ |
|---|---|---|---|---|
| Decision Tree | 0.9716 (± .0104) | 0.9723 (± .0106) | 0.9678 (± .0119) | 0.9697 (± .0110) |
| CNN | 0.9349 (± .0115) | 0.9298 (± .0131) | 0.9308 (± .0123) | 0.9377 (± .0123) |
| MobileNet V2 | 0.9743 (± .0046) | 0.9726 (± .0046) | 0.9717 (± .0054) | 0.9718 (± .0049) |
| ViT | 0.9967 (± .0015) | 0.9960 (± .0020) | 0.9966 (± .0017) | 0.9962 (± .0018) |
| Human Performance | 0.7500 (± .0589) | 0.7588 (± .0453) | 0.7500 (± .0589) | 0.7519 (± .0524) |

Table 1: Macro-averaged performance metrics for the models averaged over ten random seeds (standard deviation in brackets).

them through a user study involving the stakeholders for the task of banana ripeness classification in a real fruit market.

## 5.1 PERFORMANCE

To measure the ability of our selected models to produce correct predictions, we resort to commonly used classification metrics: accuracy, macro-averaged precision, macro-averaged recall, and macro-averaged F1-score. All methods are tested using 5-fold cross-validation, repeated ten times with different random seeds to strengthen the results.

Table 1 showcases the results achieved with both deep-learning methods and the DT. We additionally report the human performance, which is the average of the scores obtained by three stakeholders on the classification of a balanced dataset of 300 randomly sampled images from the original non-augmented dataset ($\sim 20\%$). It is easy to see that all methods achieve excellent results, with all metrics surpassing the 90s percentile scores and improving on a human baseline. It is worth remembering that these results are achieved on the datasets augmented with images that have gone through various augmentations, which makes them more robust at the cost of small decreases in performance. Further detailed in the supplementary material, error analysis reveals that mistakes always occur because the classifiers select an adjacent class (*e.g.*, class 2 instead of 1).

The ViT model achieves a near-perfect score among the selected methods for all metrics. The DT also obtained outstanding results, though this required comparatively more effort (including the standardisation of the luminance and the extensive grid search). Nevertheless, this process allows the DT to have results comparable to those of MobileNetV2.

## 5.2 EXPLAINABILITY

We compare the SHAP explanations for the DL models with the handcrafted explanations based on RGB colour designed for the DT. Fig. 2 and 3 compare the two types of explanations for the same input. It is easy to see that the masks produced by SHAP do not highlight meaningful features of the image. Indeed, we can observe that the regions highlighted are apparently random. Not only that, in our case, SHAP's visualisation for the CNN always presented the same result for all classes, seemingly valuing features for grade 4 highly (even when the CNN correctly classified other ripeness stages). The situation does not change when we visually examine the explanations generated by the methods throughout the whole dataset. This does not necessarily mean that the explanations generated by SHAP are not faithful to the model's inner workings. Rather, our intuitive interpretation of the highlighted regions is misaligned with how the model uses those features internally. As such, we can only conclude that, despite their plausibility, these visualisations are inadequate as significant explanations.

Conversely, our explanation for the DT is faithful to the model's inner workings by design. This strategy provides the user with a much more informative explanation that is intuitively understandable, plausible and faithful to how the model works. An ad hoc user study confirms such results.

### USER STUDY

We designed a user study to investigate the users' preferences about the generated explanations for the model predictions. The users involved in the study are stakeholders in the grading of banana

ripeness, consisting of 20 people with different backgrounds and expertise with artificial intelligence tools. We submitted an online questionnaire to each user. The complete questionnaire is reported in the supplementary material. The questionnaire introduces the task and asks the users to compare two types of explanations for the same input and prediction: (i) the mask generated by SHAP and (ii) the representation of the input colour in the RGB gamut. Explanations (i) pertain to the ViT model (the best-performing one), while explanation (ii) is generated from the DT. The object of the comparison is how much the proposed explanation allows you to answer why the model made that prediction.

When asked about the importance of explaining the model's behaviour, all participants believed that an associated explanation is somewhat necessary, with most thinking it to be essential. As for the preferred explanation method, ten out of twenty respondents considered the RGB gamut area produced by the DT to be the most effective, eight voted for the SHAP heatmap explanation, and three declared that no explanation was helpful to them [2]. This result is certainly interesting; though SHAP's visualisations do not provide an unambiguous explanation, their visual nature was still enough to make half of the participants deem them trustworthy in conveying why the prediction was made. Finally, 80% of respondents declared that the chosen explanation would improve their trust in the model, and 70% are ready to trade about 5% of the classifier accuracy for a more transparent and human-explainable decision process. Considering that the accuracy loss between the DT and the most accurate model is only around 2.5% for our classification task and well above human performance, there appears to be little reason to prefer the latter to the more explainable one. We report the complete results of our study as supplementary material.

## 6 FUTURE WORK

Using simple classifiers on a few manually extracted features can be much more problematic on more complex tasks, as this could severely limit the performance of the models. Indeed, we do not make the point that more transparent models should *always* be used; many cognitive tasks would be nearly impossible without the progress obtained through DL. For this specific task, we selected a simple strategy to provide an intuitive explanation to non-ML-expert users based on the average colour of the whole image. This can be refined iteratively to incorporate more complex features while accounting for explainability. We plan to explore strategies to serve explanations using higher numbers of features, for example, considering the pixel colour distribution. Moreover, in line with the *explainability by design* principle, we plan to research the usage of regularisation strategies to improve the explainability of complex DL models. This topic has already been explored (Wu et al., 2018), mostly tackling the problem of robustness, which has indeed been linked to the issue of explainability (Ross & Doshi-Velez, 2018). It would be interesting to explore whether and how adding constraints on the features extracted by NNs could help produce more understandable explanations by the end-users.

## 7 CONCLUSIONS

This paper discusses the explainability of ML models by providing high-level guidelines to tackle ML problems. As an example, we compare three DL models to a DT for classifying bananas into four ripeness stages. While the DT leads to slightly lower accuracy scores, it produces much more interpretable results. This task showcases how an intuitive explanation strategy can be devised by *model design* rather than with a *post-hoc* approach. We argue that working with a more transparent model and stakeholder-understandable features, where possible, can allow for satisfactory explanations with minimal loss in accuracy. To validate our claim, we conducted a pilot user study on 20 users, comparing the explanations produced by SHAP, a popular model-agnostic explainability method for DL models, against those produced by combining colour features and DT rule interpretations. The study results indicate users' tendency to accept minor accuracy losses, favouring a more understandable model. However, they also showcase how non-expert users prefer more straightforward explanations, regardless of whether they are well-founded.

---

[2]One participant selected both the RGB explanation and the "neither" option.

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
