# Supplementary Materials

Stop overkilling simple tasks with black-box models: use transparent models instead

## Additional considerations

### Decision Tree feature extraction

In this work, we argued for a Decision Tree based on a restricted number of features, *i.e.*, the three per-channel average color values of the pictures in the RGB color space. It could be argued that methods such as Principal Component Analysis (PCA) could be used to down-project high dimensional feature spaces for visualization, thus allowing for a larger number of features. However, we argue that the cognitive load of a user interacting with such an explanation increases exponentially with the number of selected features, regardless of down-scaling. Our aim is to mimic, features-wise, the discriminating factors that a human would identify for classifying the ripeness stage of bananas, thus providing an explanation that satisfies the user's intuition while staying true to the inner decision-making of the model.

### Training details

The baseline CNN model was trained with a batch size of 64, while pre-trained models used a batch size of 16. In all cases, the initial learning rate was set to $1.5e^{-5}$, and patience for early stopping was set to 2 epochs. All the neural models converged relatively fast (around 13 epochs for the CNN, 10 for MobileNetV2, and 5 for ViT). As per the DT, an extensive 5-fold grid search determined the best parameters to be the entropy criterion, random best node splitting, and minimal cost-complexity pruning alpha parameter set to 0.0016. The luminance pre-processing step sets the Y channel to 0.8 for all images.

The MobileNetV2 model has been pre-trained on ImageNet-1k (1.3 million images, 1000 classes, while the ViT model has been pre-trained on ImageNet-21k (14 million images, 21,843 classes).

### Results for other methods

| | Results averaged over 10 random seeds - RGB - 20% test split | | | | | | | |
| --- | --- | --- | --- | --- | --- | --- | --- | --- |
| | Accuracy | | Precision | | Recall | | F1 | |
| | avg | std | avg | std | avg | std | avg | std |
| **Linear SVM** | .8812 | .0457 | .9084 | .0265 | .8574 | .0461 | .8676 | .0501 |
| **Naive Bayes (multinomial)** | .8518 | .0209 | .8441 | .0211 | .8424 | .0195 | .8421 | .0196 |
| **SVM (poly kernel, degree 8)** | .9762 | .0071 | .9761 | .0063 | .9734 | .0084 | .9746 | .0074 |

# Additional parameters and results

| Ripeness stage | 1 | 2 | 3 | 4 |
|---|---|---|---|---|
| # original samples | 164 | 266 | 286 | 211 |
| # augmented samples | 244 | 399 | 438 | 325 |

Table 1: Number of sample images per ripeness value in the dataset, both in its original and augmented versions.

| Transform | Parameters |
|---|---|
| Rotation | Up to 270° |
| Random affine | $d \in [0, 70]°, t \in [0.1, 0.3], s \in [0.7, 0.9]$ |
| Elastic transform | $\alpha = 80.0$ |
| Random crop | $128x128$ window |
| Gaussian blur | kernel size $\in [5, 9], \sigma \in [0.1, 2]$ |
| Random erasing | $s \in [0.02, 0.15]$ |
| Random perspective | distortion scale $= 0.5$ |

Table 2: Parameters for augmentations performed statically on the dataset. d = degrees, t = translate, s = scale.

| | Inference time (on CPU, ms) | Inference time (on GPU, ms) | Average model size |
|---|---|---|---|
| Decision Tree | 0.0008 (± 2e−5) | N/A | N/A |
| CNN | 0.1046 (± 0.0039) | 0.0023 (± 0.0101) | ~ 828 MB |
| MobileNet V2 | 0.0289 (± 0.0019) | 0.0079 (± 0.0102) | ~ 14 MB |
| ViT | 0.3309 (± 0.0080) | 0.0101 (± 0.0106) | ~ 343 MB |

Table 3: Portability results for the various models in terms of inference time and average model size on disk.

# Image augmentation and preprocessing

We show an example of the transformation performed on images before classification. The additional "Luminance Normalization" step is only carried out for the input of the Decision Tree.

### Original Image (square ratio)

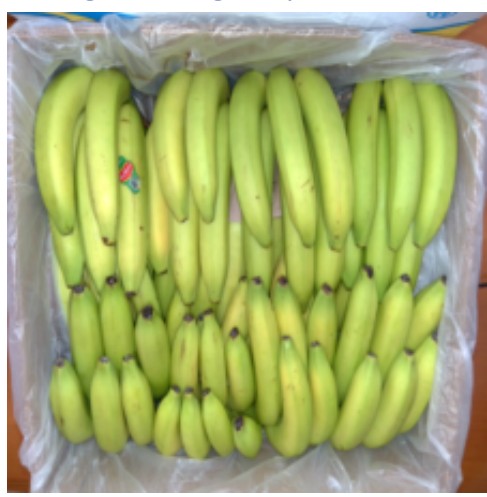

### Segmented image

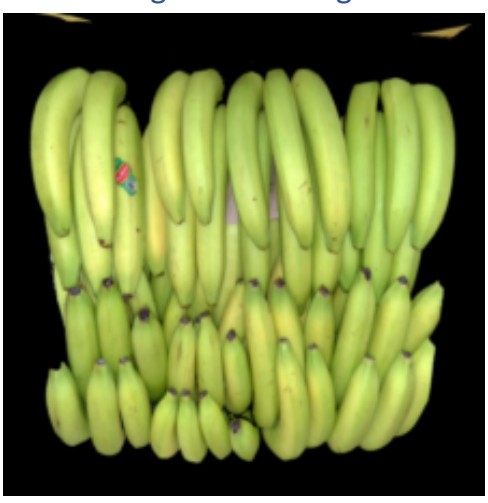

### Segmented + Luminance Normalization

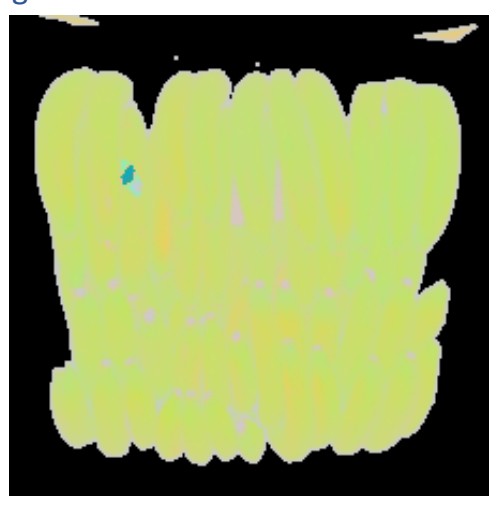

### Segmented + Augmentation (Rotate, Blur, Flip)

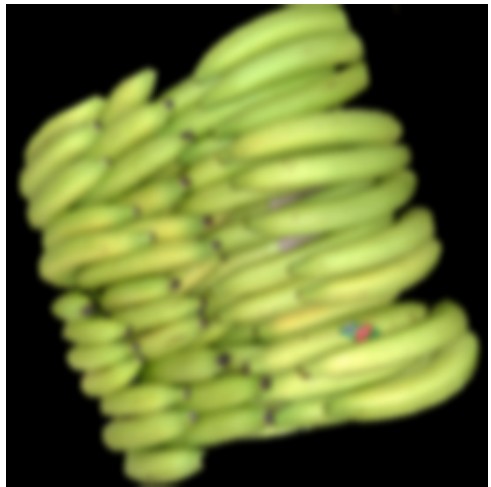

# Error analysis

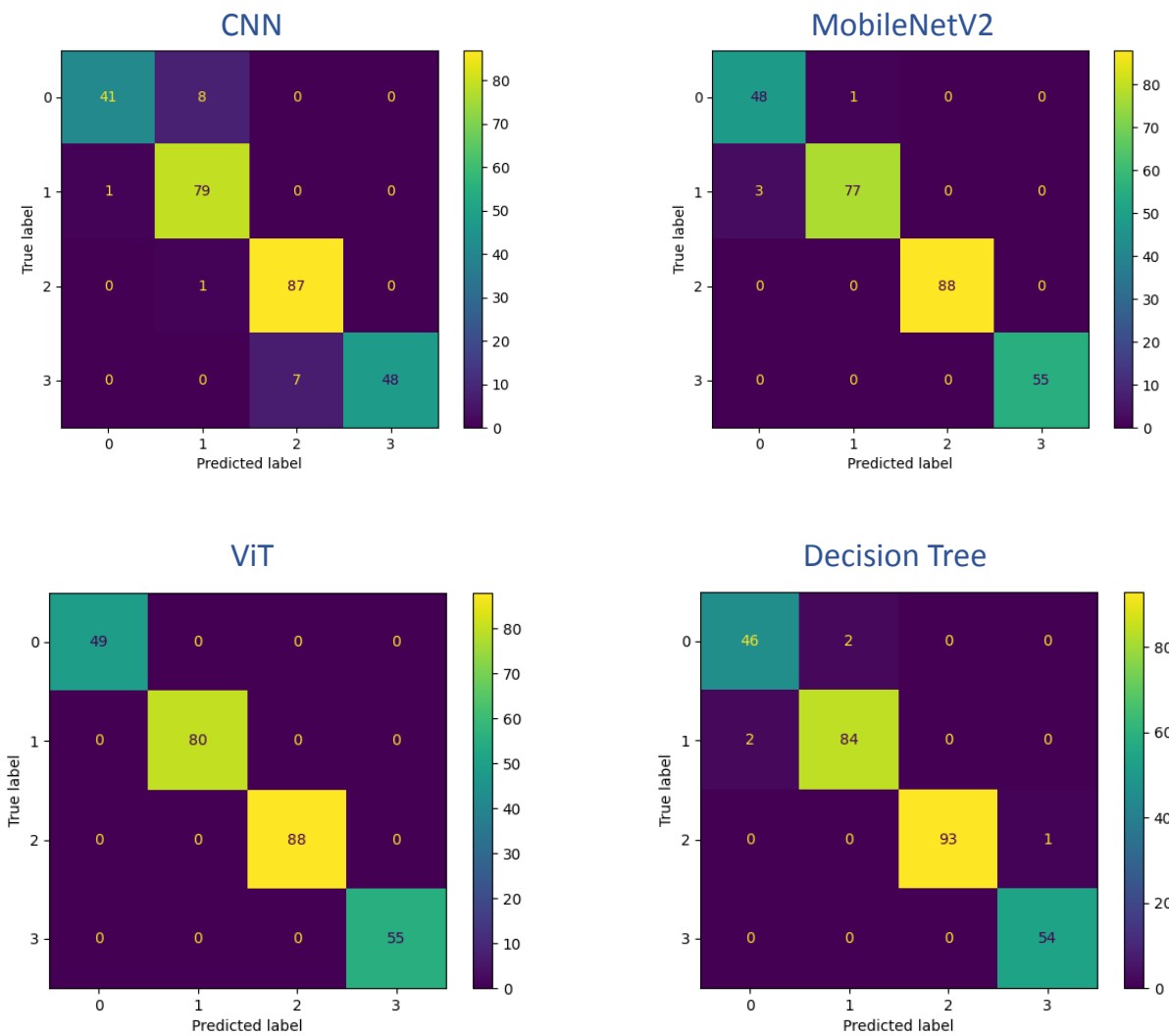

Simple visual inspection of the confusion matrices for the models reveal that mistakes always fall into adjacent categories (close to the diagonal). Moreover, it is apparent that it is easier to mistake class 1 for 2 and class 3 for 4. This is likely to be because classes 1 and 2 are similarly green-tinted, while classes 3 and 4 appear to be much more on the yellow-brownish side.

The reported example for the ViT model presents perfect performance; throughout all folds and repetitions, this was almost always the case, though the model would sometimes make a few mistakes (similar to the ones just explained).

# Explanation examples

We report different visualizations obtained to explain the prediction over 4 different images using the Decision Tree color visualization, and the ViT model with SHAP.

## SHAP

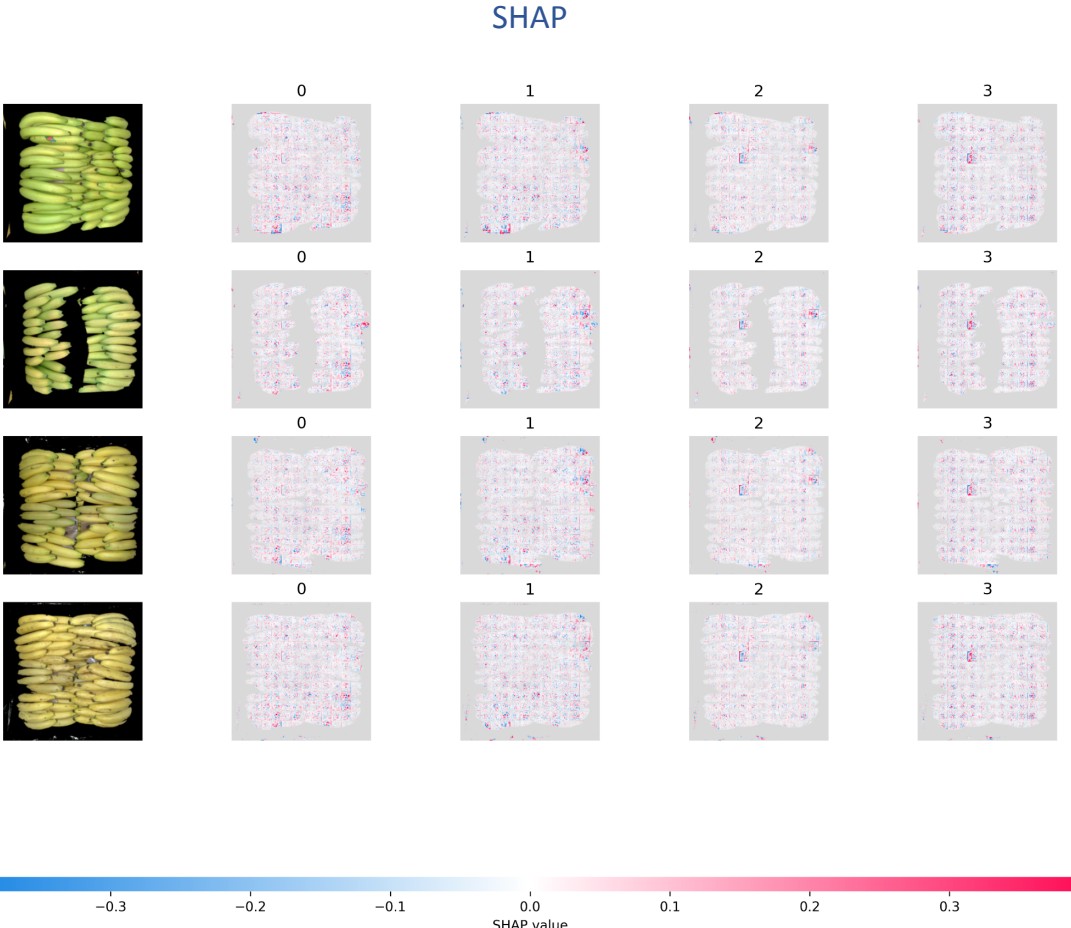

# Decision Tree

### Ripeness area for value 0

### Ripeness area for value 1

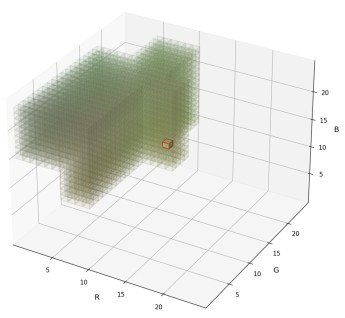

### Ripeness area for value 2

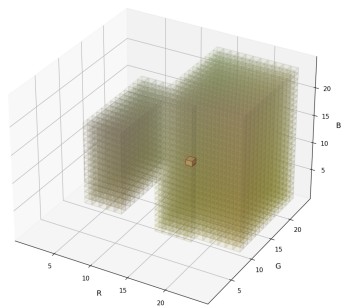

### Ripeness area for value 3

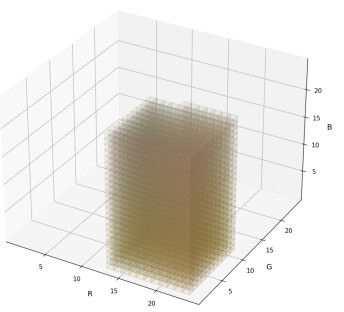

# Detailed user study

Note that the form was originally written in Italian as it was designed for an Italian audience. We report here the English translation of the questions and results.

## Form template

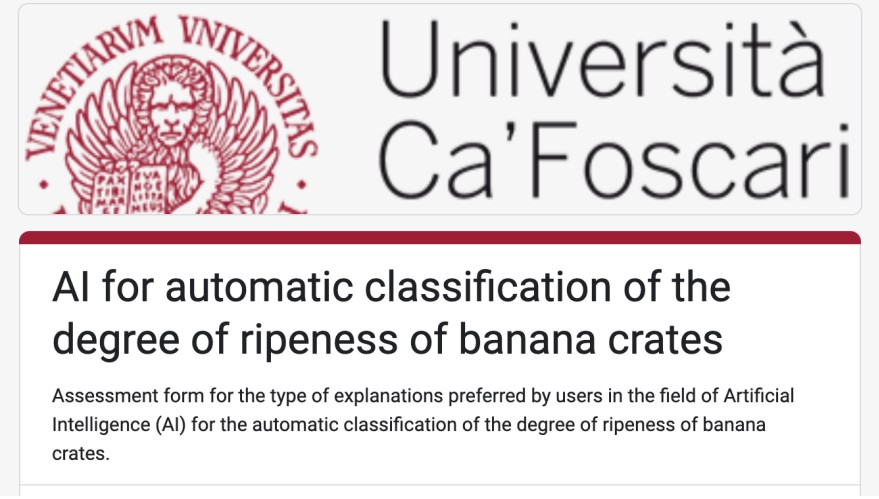

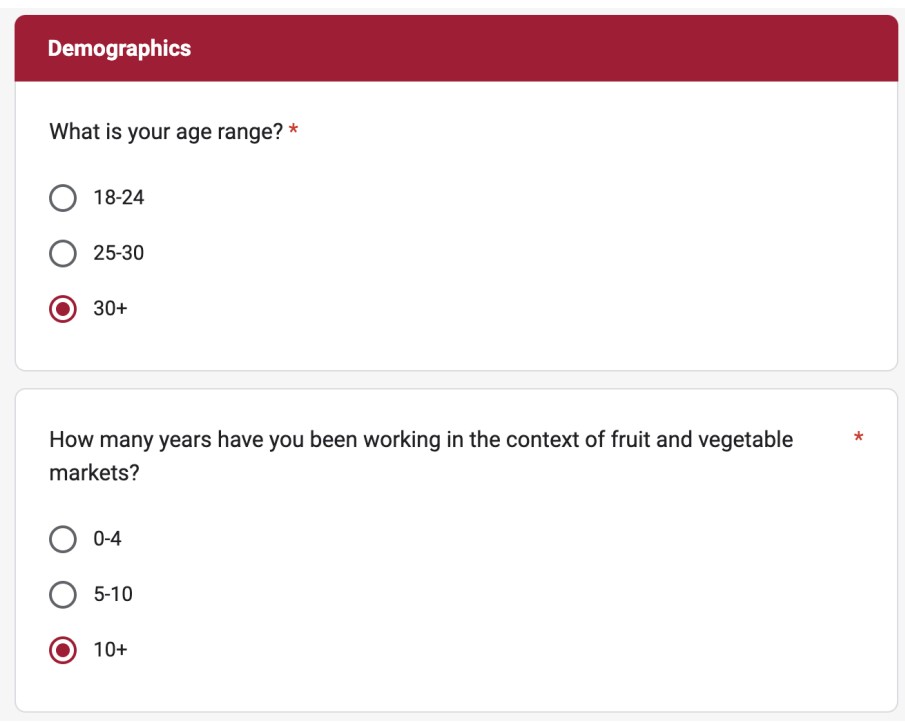

## Relationship with Artificial Intelligence (AI)

How familiar are you with the concept of Artificial Intelligence (AI)? *

|  | 1 | 2 | 3 | 4 | 5 | 6 | 7 |  |
|---|---|---|---|---|---|---|---|---|
| Never heard of it | ◉ | ○ | ○ | ○ | ○ | ○ | ○ | I have worked with AI methods and know the basics |

An artificial intelligence method makes predictions for input data so as to automate a decision-making process. How much would you rely on an AI-generated prediction? *

|  | 1 | 2 | 3 | 4 | 5 | 6 | 7 |  |
|---|---|---|---|---|---|---|---|---|
| No trust at all | ◉ | ○ | ○ | ○ | ○ | ○ | ○ | Complete trust in the AI system |

Although an AI model can generate very accurate predictions, for some models it is not always possible to explain what mechanism produced those explanations. How important is it for you to receive an explanation associated with predicting an AI method? *

|  | 1 | 2 | 3 | 4 | 5 | 6 | 7 |  |
|---|---|---|---|---|---|---|---|---|
| Not at all important, I trust the model completely | ◉ | ○ | ○ | ○ | ○ | ○ | ○ | It is of utmost importance, I would never trust an AI method that doesn't provide an explanation |

## Introduction to the banana ripeness classification task

Consider the following ripeness of bananas, from underripe (1) to very ripe (4).

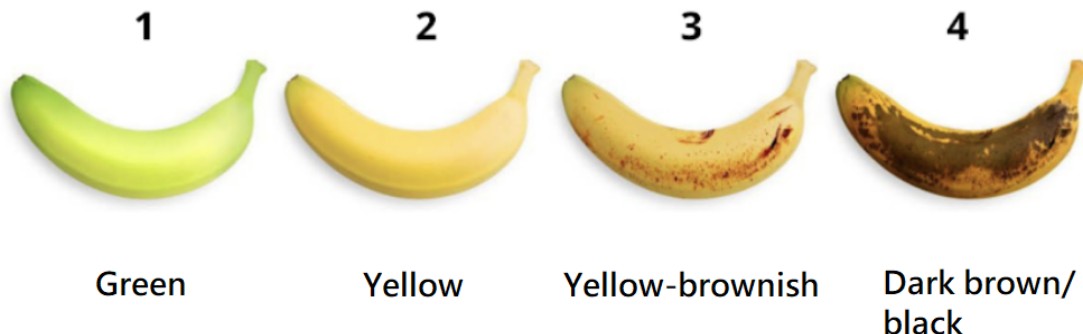

| 1 | 2 | 3 | 4 |
|---|---|---|---|
| Green | Yellow | Yellow-brownish | Dark brown/black |

Imagine an AI-based classifier capable of accepting as input a photo of a box of bananas of the same degree of ripeness and outputting an estimate of the degree of ripeness of the entire box from 1 (not very ripe) to 4 (very ripe), as in the example.

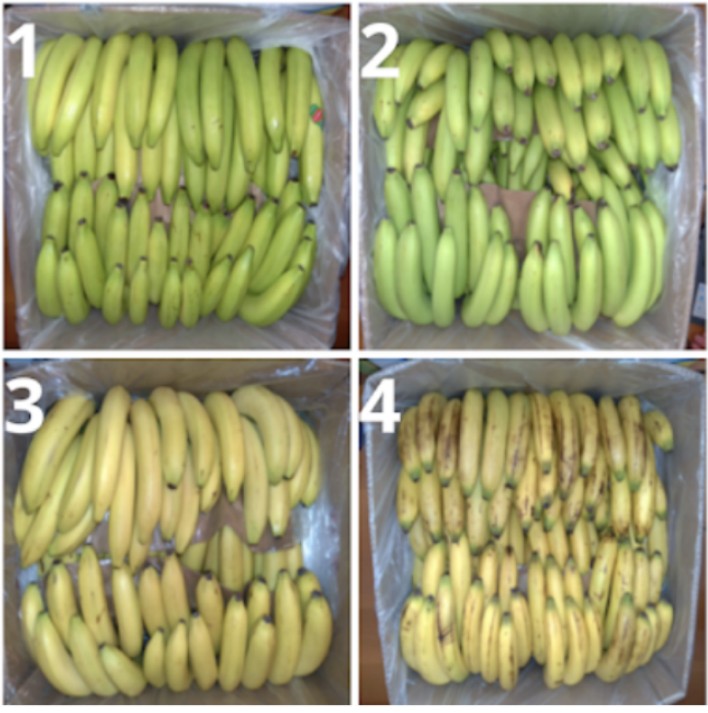

# Explanation of AI prediction

The following **two** sections represent as many explanations for the prediction of an AI-based classifier.

## Explanation 1

The animation shows the workflow from the original input image to the generation of an explanation of the classifier's prediction. Background is removed for ease of classification. Classes 1 to 4 represent the degrees of ripeness of the banana crates.

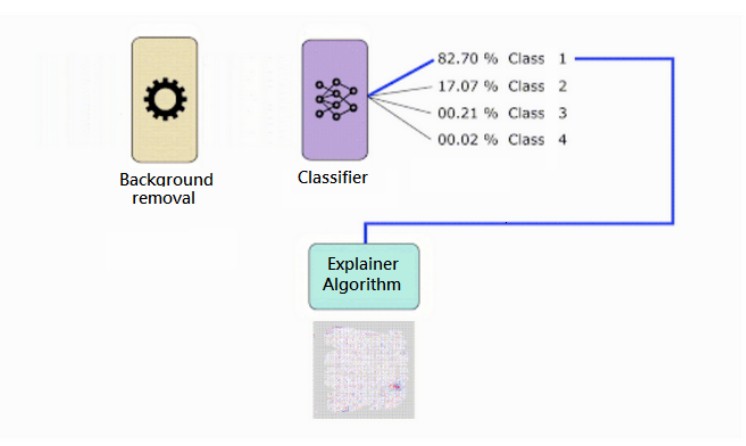

The parts of the input image highlighted in red contributed positively to the prediction, those highlighted in blue contributed negatively.

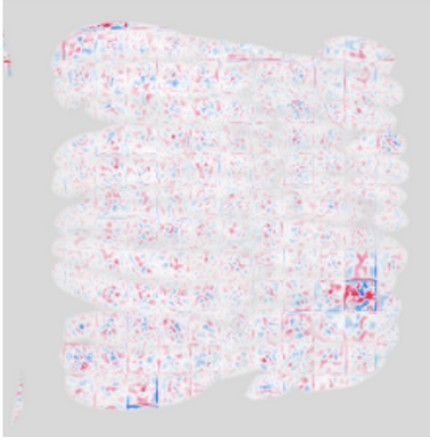

**Premise**

In this case the classifier is completely based on the average color of the input image. A simple representation of colors is RGB (i.e., R(ed, red), G(reen - green), B(lue, blue)). By combining the colors red, green and blue it is possible to obtain any other shade of color. The set of representable colors is a cube where it is possible to identify a specific color with its RGB coordinates (e.g., R=0.4, G=0.7, B=0.1).

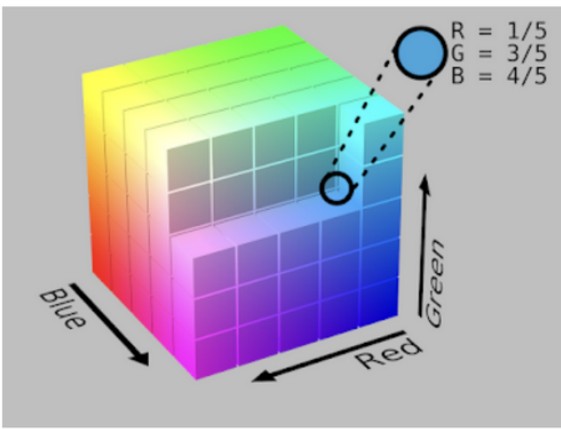

**Explanation**

In this case the classifier learns to associate the degrees of maturation to a set of RGB colors. Specifically, learn which shades of green correspond to low degrees of ripeness, and which shades of yellow/brown correspond to higher degrees of ripeness.

From each input image the background is removed and the average color is extracted. Based on the average color alone, the classifier predicts the associated degree of ripeness.

Each of the four graphs in the following diagram corresponds to a set of colors (identified by as many cubes), which define a portion of the RGB space. The classifier learns to associate each set of RGB colors with one of the four maturation classes.

Note how grades 1 and 2 are associated with greenish colors. Grades 3 and 4, on the other hand, have shades tending towards yellow/brown.

The input image in the example has degree of maturation 1: as highlighted in the image outlined in red, its average color (represented by the cube in the red circle) correctly falls within the color area that the classifier associates with degree 1.

Similarly, the same color appears outside the areas associated with grades 2, 3 and 4: the same cube is colored black when it is outside the set of colors relevant to these ripening grades.

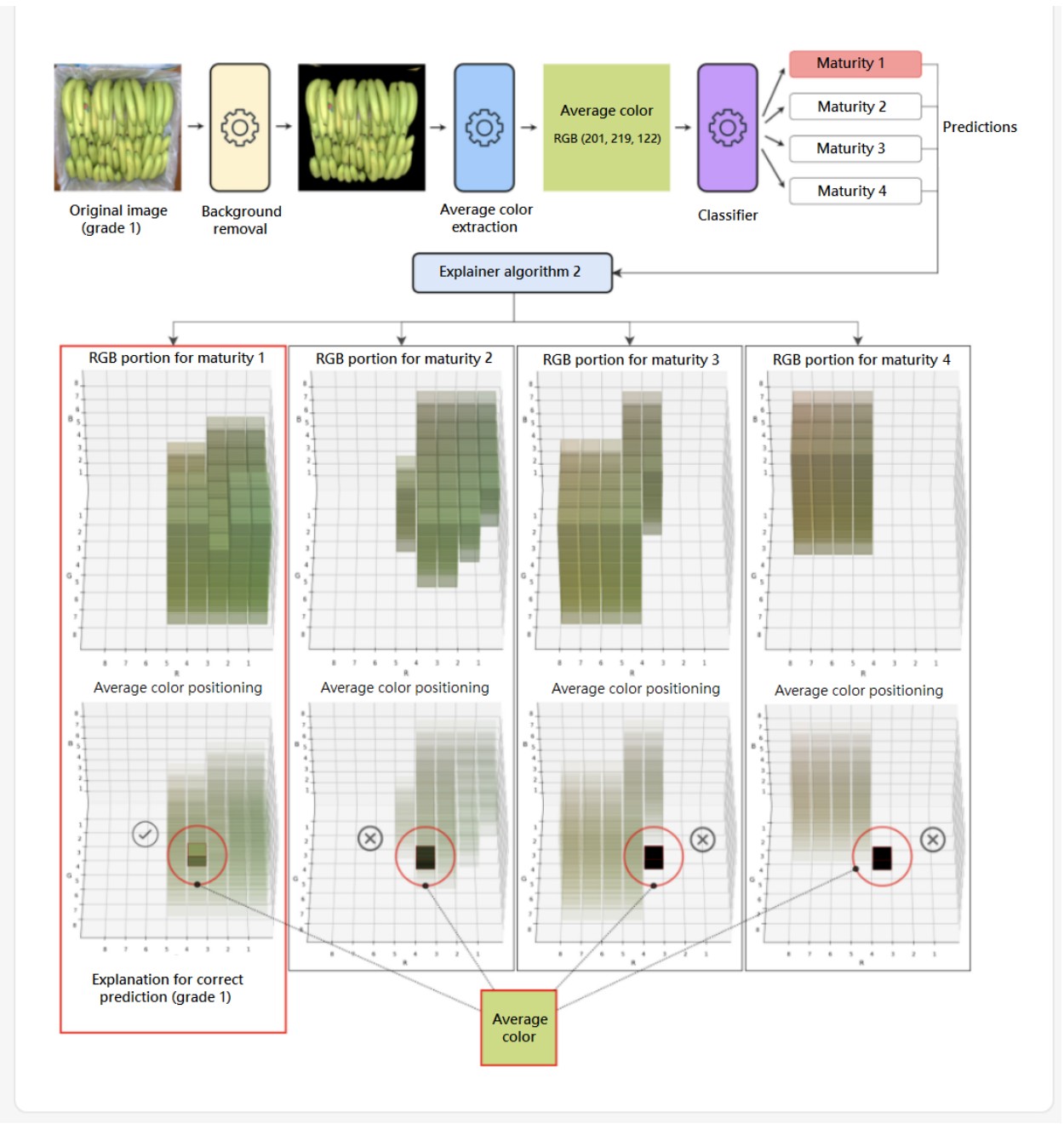

## Preferred explanation

Which of the three explanations of the prediction of an AI-based classifier do you *
find most useful for understandability and informativeness?

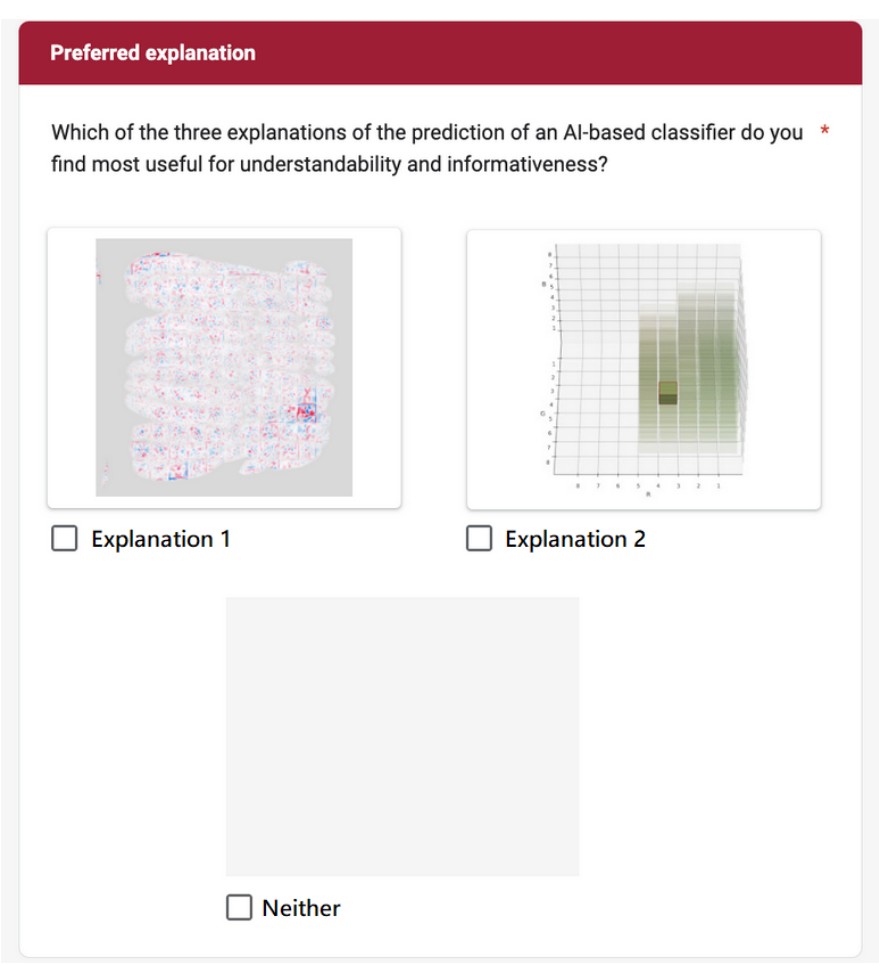

☐ Explanation 1

☐ Explanation 2

☐ Neither

## Results

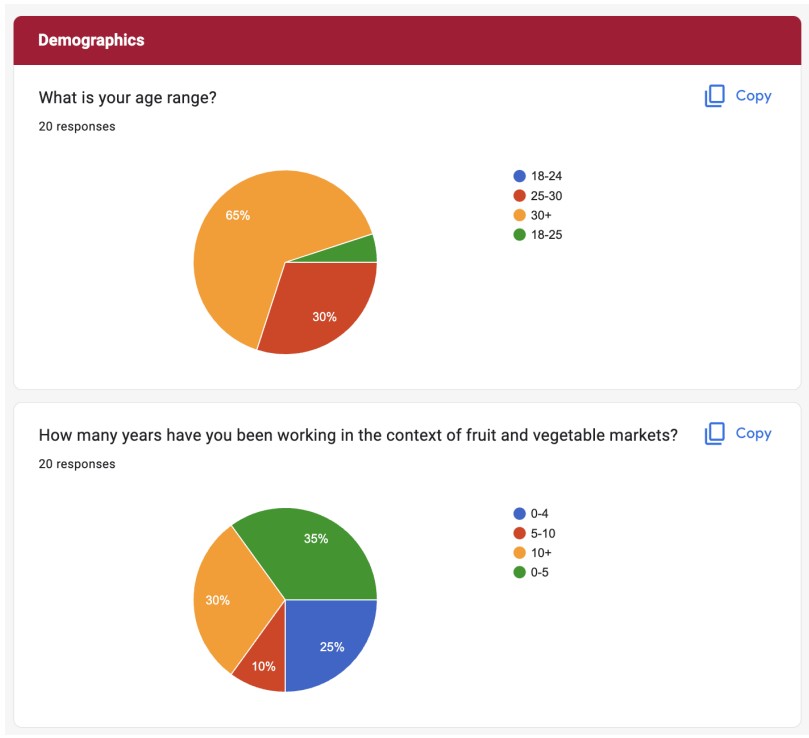

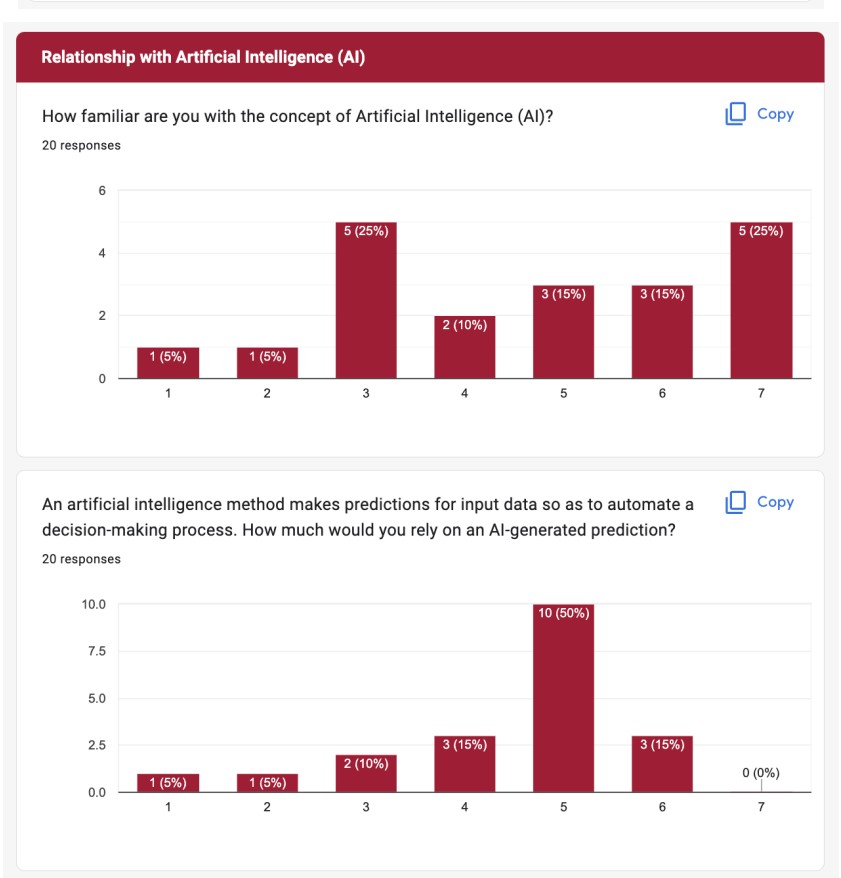

## Preferred explanation

Which of the three explanations of the prediction of an AI-based classifier do you find most useful for understandability and informativeness?      📋 Copy

20 responses

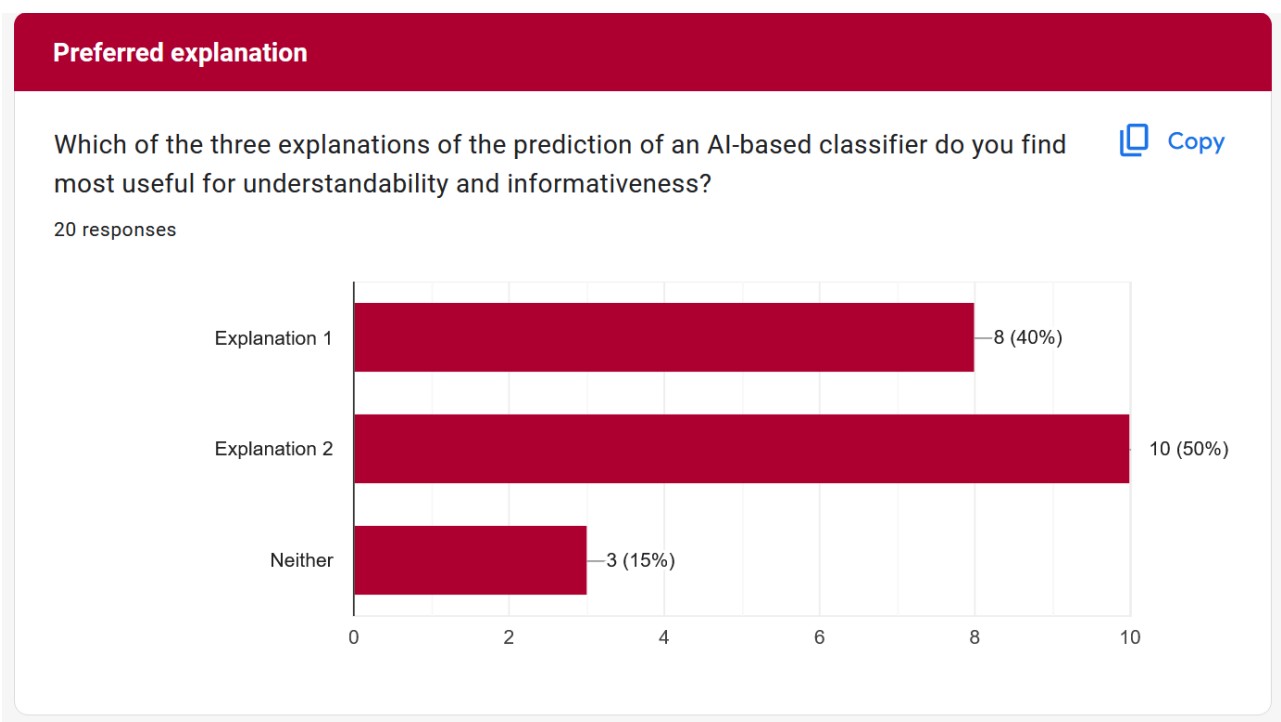

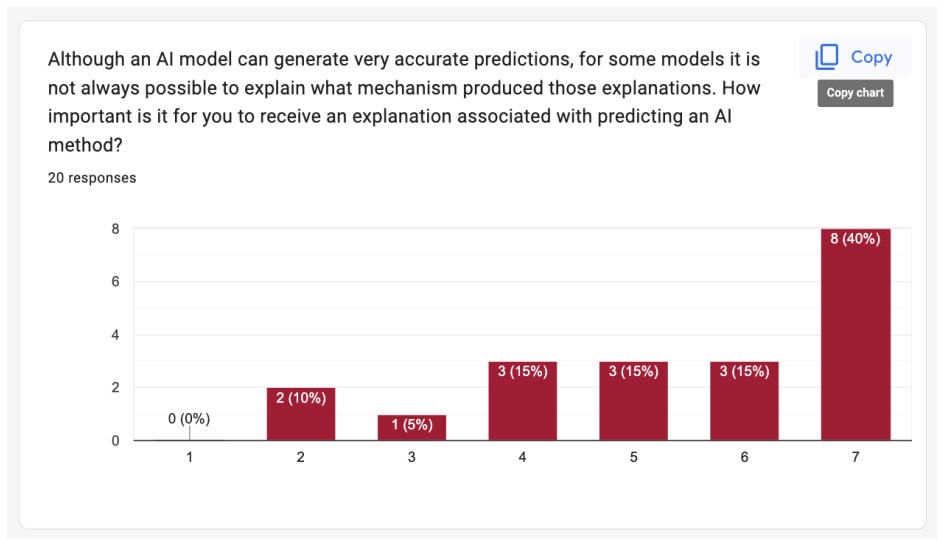

Although an AI model can generate very accurate predictions, for some models it is not always possible to explain what mechanism produced those explanations. How important is it for you to receive an explanation associated with predicting an AI method?

20 responses

Does the presence of the preferred explanations in the previous point increase your confidence in the AI-based classifier?

20 responses

- Yes
- No
- Neither was understandable

80%
10%
10%

Would you be willing to use a slightly less accurate classifier (about 5% less) but which provides more informative explanations?

20 responses

- Yes
- No

70%
30%