# OpenReview forum: "Stop overkilling simple tasks with black-box models, use more transparent models instead"
_ICLR.cc/2024/Conference — ICLR 2024 Conference Withdrawn Submission_

### Official Review · Reviewer_wzax · 2023-10-25

**Soundness:** 1 poor
**Presentation:** 1 poor
**Contribution:** 1 poor
**Rating:** 3
**Confidence:** 4

**Summary:**

The paper presents a case study where the performance of a DT is compared to those of three DL approaches to classify the ripeness of banana crates.

**Strengths:**

+ Interesting topic

**Weaknesses:**

- Results are not generalizable and are not statistically significant nor for the performance, nor for the user study.
- The user study is not sufficiently detailed in the main paper as for this type of contribution is a central part. Thus, there is a lack of a central contribution.
- At least a comparison against ProtoPNet of Rudin would have been necessary, if not against other baseline approaches such as a simple kNN.

**Questions:**

- The recommendation for the authors for this kind of contribution is to change venue.

---

### Official Review · Reviewer_qoq3 · 2023-10-27

**Soundness:** 3 good
**Presentation:** 3 good
**Contribution:** 2 fair
**Rating:** 3
**Confidence:** 5

**Summary:**

The thesis of this paper is that simple machine learning tasks should be addressed with simple, interpretable or explainable models, rather than with more complex (but also less transparent) deep learning approaches, even at the cost of a slightly lower performance. The paper makes the case of an image classification task, where simple color features and decision trees produce transparent and intuitive solutions that are only marginally worse than state-of-the-art deep architectures.

**Strengths:**

+ I totally sympathize with the claim of the paper, since I agree that simpler methods should be preferred whenever possible, i.e., when there is no significant performance gap with respect to more complex (e.g., deep learning) models and where explainability is a crucial need.

+ The paper is very well written, clear, and easy to follow.

**Weaknesses:**

- The thesis of the paper is certainly not novel. The paper itself cites the work by Cynthia Rudin, but other works have recently proved the advantages of classical approaches like decision trees with respect to more complex deep learning approaches: e.g., see Grinsztajn et al., "Why do tree-based models still outperform deep learning on typical tabular data?", NeurIPS 2022 and references therein.

- The paper only presents a case study on a single (image processing problem) which allows a solution based on very simple features (colors extracted from RGB). Therefore, it is vert hard to assess to what extent the methodology could be generalized, in fact the chosen features are in this case highly intuitiveand easy to understand for humans, and the explanation representation based on the RGB space depends on such features. It is not clear which other problems and tasks could present a similar scenario.

- The paper looks in this sense more suited for a computer-vision workshop or conference, since the thesis is not novel, and the methodology is tailored to the case study. I am not sure how many real-world problems in computer vision fall in this category of tasks that can be easily solved by a decision tree (or a similar, simple technique). A study on this would be extremely interesting.

**Questions:**

* The proposed methodology exploits a pre-processing stage that transforms images from RGB to YUV and back to RGB after a luminance adjustment. It is not clear from the paper, but I think that also the deep learning approaches tested in the paper should work on the same images, processed with the same technique, to make a fair comparison.

* On pag. 7, the paper states that "for every non-leaf node, the DT learns a threshold value for one of its given features, thus producing two children" -> This is true for the task considered in the paper, but not in general, where also discrete variables with multiple outcomes could be used as features (thus requiring no thresholds while producing multiple children).

---

- Pag. 2, "for whom strategies" -> "for which strategies"
- Pag. 2, "that are more intuitive" -> "that are most intuitive"
- Pag. 4, "while the human performance" -> "while human performance"

---

### Official Review · Reviewer_tBTh · 2023-11-01

**Soundness:** 2 fair
**Presentation:** 2 fair
**Contribution:** 1 poor
**Rating:** 5
**Confidence:** 5

**Summary:**

The authors advocate that sometimes, using deep learning models for simple machine learning tasks is overkill. In particular, they work on a use case that requires the classification of fruit ripeness (on bananas) and compare several deep-learning models to a decision tree. They compare the models based on their discriminative performance and perform some assessments regarding how explainable such models are. Furthermore, they propose a visualization to assess which factors influenced the decision tree prediction.

**Strengths:**

- the authors propose a novel visualization method that provides a degree of explainability to machine learning models where color is of key importance to predict a particular outcome.
 - the paper is clearly structured
 - the authors tackle a relevant topic and work on a real-world use case

**Weaknesses:**

- the quality of the paper is average. The survey conducted to evaluate the quality of explanations (a) lacks the depth to understand why people prefer a particular explanation, (b) compares the proposed method only against one explainability method, (c) both methods convey different information, and, thus, it isn't easy to assess whether the different perceptions relate to (i) the information conveyed, (ii) how the information is conveyed; and (d) most of the results related to explainable artificial intelligence are not presented in the paper (but the supplementary material).
 - a more extensive experimental setup would strengthen their claims. In particular, they should consider other explainability methods.

**Questions:**

We consider the paper interesting. While the claims lack novelty, the fact that it addresses a real-world use case makes it relevant. Nevertheless, we would like direct your attention to the following improvement opportunities:

 - 4.3 - Explainability strategy: (a) could the proposed plot be enhanced, e.g., by rendering a prototype/the average/expected values, so that they serve to contrast against the actual values they provide in the plot for a particular model? (showing the average value within the same plot / not another plot nearby), (b) can a metric be derived by comparing the actual plot vs. the one that displays the average value for the class? Such a metric would be a great contribution, given it would summarize the visual assessment in a number, making it easily comparable across other cases too.
 - Experiments:
 	- The authors should make more clear the experiments performed, the goals of the experiments, the metrics considered, and the results obtained.
 	- Most of the results reported by the authors relate to the discriminative power of the machine learning models, but very little insight is provided regarding the explainable artificial intelligence component - the key aspect of the paper. We encourage the authors to restructure this section and the whole paper so that it better reflects their contribution. The authors could provide better insights by highlighting: e.g., whether the participant's background influenced the response outcomes / XAI perception, (b) are the differences among results significant / can definitive conclusions be drawn from the limited sample?. Some aspects not considered in the surveys, but that would provide value to the research, could be e.g., (i) what aspects of the visualization helps better understand the predictions and what additional aspects should be considered?, (ii) does displaying average values help the user to understand a particular prediction?, (iii) why did the people not perceive the Shapley explanation as useful? e.g. overlaying the Shapley dots on the bananas image would enhance perception?, (iv) Shapley and the proposed method convey different information to the user. Preference of one against the other one relates to the information conveyed or how such information is conveyed?, (v) people who were not convinced by either explanation - can provide some additional insights on a follow up interview? It would be interesting to understand why neither of the explanations is compelling / informative to them. (vi) It would be great to understand how such explanations (may) affect the decision-making process, and, therefore, generate direct impact.

---

### Official Review · Reviewer_92wc · 2023-11-01

**Soundness:** 2 fair
**Presentation:** 2 fair
**Contribution:** 1 poor
**Rating:** 3
**Confidence:** 3

**Summary:**

This paper addresses the problem of opacity in the decision making process of deep networks, and investigates the use of simpler, more transparent models for simple tasks that can lead to improved explainability without a significant cost in accuracy. Considering a task of classifying stages of banana ripeness, it compares and discusses the performance and explainability of different traditional and deep learning methods.

**Strengths:**

1. The problem is well motivated and highly relevant. The black-box nature of deep networks is a significant concern particularly when used in safety critical applications, and it is crucial to carefully balance the need for performance with that of being able to trust the model's decisions. In this context, using more interpretable models where possible could be very useful. This is particularly relevant given that post-hoc explanations of deep networks have often shown to be misleading.
1. For the specific task of classification of banana ripeness, this work does a fairly detailed study on the tradeoff between performance and explainability.

**Weaknesses:**

1. Evaluation is performed on a single four class classification task for banana ripeness, which appears to be too simple to draw broad conclusions from. It is unclear what insights gained from this work that can be more broadly applied to other more complex tasks. All models achieve very high accuracies, which would not be the case in general. The size of the dataset used is also very small.
1. Using different inputs for the decision tree (DT) model (which uses a three dimensional input consisting of mean (R,G,B) values) and the deep learning (DL) models (which use raw images) does not seem to allow a fair comparison between their explanations. The use of mean RGB values as features for DT significantly simplies the task, and using the same inputs for DL models would also likely give highly interpretable explanations. As such, in this case the explanations from DT seem to be more interpretable primarily due to the input format used, without necessarily pointing to an advantage of DT over DL methods for such a task.
1. The explanations are evaluated only with a user study, which asks users for their preferred explanation. However, this would only evaluate the plausibility of explanations, and not their faithfulness. Relying only on such a study could favour explanations that "look" reasonable even if they are highly misleading about the model's working. A more thorough evaluation including for faithfulness appears to be necessary.

**Questions:**

Clarifications on the points raised under Weaknesses would be helpful.